# Demagnetization Fault Detection and Location in PMSM Based on Correlation Coefficient of Branch Current Signals

**Yinquan Yu** [1,2,3,*], **Haixi Gao** [1,2,3], **Qiping Chen** [1,2], **Peng Liu** [4] and **Shuangxia Niu** [5]

[1] School of Mechatronics and Vehicle Engineering, East China Jiaotong University, Nanchang 330013, China; ghx18397991732@gmail.com (H.G.); 2758@ecjtu.edu.cn (Q.C.)

[2] Key Laboratory of Conveyance and Equipment of Ministry of Education, East China Jiaotong University, Nanchang 330013, China

[3] Institute of Precision Machining and Intelligent Equipment Manufacturing, East China Jiaotong University, Nanchang 330013, China

[4] School of Electrical and Automation Engineering, East China Jiaotong University, Nanchang 330013, China; 3132@ecjtu.edu.cn

[5] Department of Electrical Engineering, The Hong Kong Polytechnic University, Hong Kong 999077, China; eesxniu@polyu.edu.hk

[*] Correspondence: yu_yinquan@ecjtu.edu.cn

**Abstract:** To address such challenges as an uncertain number of demagnetization poles of the permanent magnet synchronous motor (PMSM) and cases in which the fault cannot be located, this paper proposes a fault identification and location methodology based on the analysis of the motor stator current. First, the influence of the irreversible demagnetization of permanent magnets on the analytical model of the back electromotive force (Back-EMF) of the rotor in a single motor stator slot is analyzed. Moreover, considering the topology of the motor, the influence of the demagnetization fault on the stator phase current and branch current is analyzed. Since the stator phase currents cannot diagnose the partial demagnetization faults of PMSM with some topological structures, the stator branch current is selected as the signal for the identification and localization of the demagnetization fault. Secondly, the demagnetization fault diagnosis and mode recognition of the motor are carried out through the amplitude of the real-time branch current and the harmonic components of the PMSM. A sample database of demagnetization faults is established through calculation and normalization of the residual value of the stator branch current and the branch current of the healthy motor after demagnetization in one pole order. The fault threshold is obtained by analyzing the residual of the branch current of uniform demagnetization and the Pearson correlation coefficient of the fault sample database. Then, the correlation coefficient between the real-time branch current residual value of PMSM and the fault sample database is analyzed, and the number of demagnetization poles and the fault location are determined by the number and location of the calculated correlation coefficient exceeding the threshold. Finally, the feasibility and effectiveness of the proposed method are verified by the finite element analysis (FEA) results.

**Keywords:** PMSM; demagnetization fault detection and location; stator branch current; Pearson correlation coefficient; fault sample database





## 1. Introduction

The permanent magnet synchronous motor (PMSM) is widely used in multiple fields, such as sustainable energy wind power generation, new energy vehicles, and rail transport, due to its advantages of simple excitation, compact structure, and high-density power, etc. [1–3]. However, this kind of motor inevitably suffers from various malfunctions during its manufacturing, assembly, and operation. Malfunctions, e.g., rotor eccentricity, demagnetization of the permanent magnet, circuit failure, and bearing fault, would seriously undermine the reliability and security of the motor operation [4]. The demagnetization in

permanent magnet refers to the phenomenon whereby partial or uniform demagnetization occurs due to the comprehensive effect of the armature [5]. The irreversible demagnetization fault, which commonly exists in permanent magnet motors, can result in severe equipment damage and property loss if not timely identified and rectified. Therefore, it is of great importance to identify and locate demagnetization faults in PMSM [6]. The early detection of and location of demagnetization can improve the reliability and maintenance performance and prolong the service life of the motor while reducing unexpected halt and corresponding losses. More importantly, knowing the type of motor fault as well as the location and number of faulty magnetic poles can help maintenance personnel to repair the motor accurately and promptly. Therefore, it is of great significance to study demagnetization fault detection, fault location, and number of fault poles of PMSM.

At present, large quantities of research have been carried out on the diagnosis and location of demagnetization in PMSM. In these studies, modeling and experiments were adopted as the first step to extract electrical signals, such as the voltage and current of the motor, mechanical signals, such as torque and vibration, and magnetic field signals, such as magnetic density and flux. Secondly, suitable methods of processing signals were employed to extract the characteristics of faults from various signals, identify the mode, estimate the severity, and determine the position of the demagnetization. The fault of partial demagnetization is diagnosed through direct measurement of the torque, and then the specific harmonics were adopted to analyze the malfunction of demagnetization [7]. However, these above-mentioned methods can only be applied to large-sized motors, because the torque transducer is relatively expensive. Demagnetization usually affects the remanence in permanent magnets, thus further impacting the density of the radical air-gap magnetic field and the special harmonics of radical electromagnetic force [8]. When a demagnetization fault occurs in PMSM, the vibration will be so intensified that the specific vibration harmonics appear, indicating the existence of such a fault, and this can be detected by employing acceleration sensors. However, the shortcoming of the method is that the acceleration sensor is indispensable in this case. Moreover, the noises and torque signals of the PMSM are extracted by using the method of multi-sensor fusion, and the fault characteristics are extracted by multi-information fusion technology combined with wavelet transform. In this case, the demagnetization fault can be effectively detected, and the severity types of demagnetization fault can be evaluated [9]. Alternative, direct detection method of magnetic fields in PMSM is employed. First, the detection coils can be placed inside the motor. Secondly, the magnetic signals can be directly used to analyze the tooth flux of the motor as well as the inductive electromotive force of the detection coils. Then, these signals can be classified into stator component and rotor component, respectively [10,11]. Finally, the distribution of rotor components of the motor with demagnetization fault was employed for fault analysis of the motor. The aforementioned methods can effectively diagnose the demagnetization fault in the PMSM. However, these methods represent invasive diagnoses or require extra sensors, thus increasing the cost of diagnosis.

Advanced machine learning methods are also used for PMSM demagnetization fault detection. Using their advanced data processing capabilities, combined with external signals of PMSMs, such as acoustic noise, vibration, torque, etc., can non-invasively detect motor faults. An acoustic noise analysis based on a back-propagation neural network (BPNN) model was used to detect uniform demagnetization faults of PMSM [12]. Combining vibration and acoustic signal data, the data-driven early fault diagnosis method of PMSM based on Bayesian network (BN) has high diagnostic accuracy [13]. Additionally, a one-dimensional convolutional neural network (1D CNN) model analyzes the torque and current signals of the motor to diagnose the motor under a wide range of speeds, variable loads, and eccentricity effects [14]. However, these advanced algorithms require a large amount of computation and have high-end hardware requirements. Therefore, it is necessary to study a simpler method to diagnose the demagnetization fault of PMSM.

Partial demagnetization can alter the symmetry of magnetic flux inside the motor, thus triggering the back electromotive force (Back-EMF) of the motor to change. As partial

demagnetization can lead to harmonics of $f = (1 \pm n/p)$ fs of inductive electromotive force in the motor. Therefore, partial demagnetization can be diagnosed by analyzing the frequency spectra of electrical signals of the motor [15]. However, the above-mentioned harmonics components do not appear in the case of partial demagnetization if the motor is integral slot winding. The real reason for this is that, although the inductive electromotive force in a single slot after partial demagnetization has relative harmonics components, it does not appear in the stator phase current of the motor with integral slot winding because the symmetry of the motor structure leads to the canceling of them among each other. However, the motor fault can be diagnosed by using the motor zero-sequence voltage component [16]. Further analysis reveals that there are no fault harmonics in the stator phase current of PMSM with a slot-pole ratio of 3/2 and its integer multiples after a local demagnetization fault [17]. Under such circumstances, the measurement of zero-sequence voltage is the only way out in terms of detecting a loss of excitation. However, the measurement of zero-sequence voltage requires that a balanced network is connected to the neutral point of the motor, which would increase the additional cost of diagnosis. To sum up, the partial demagnetization fault can change the symmetry of magnetic flux inside the motor and lead to Back-EMF Distortion. Thus, triggering the Back-EMF of the motor can detect this fault. When detecting the state of the motor, the inductive Back-EMF of the motor cannot be measured in a direct manner from the outside of the motor. Therefore, it is generally impossible to directly use the induced Back-EMF as the signal for motor fault detection. For this reason, most researchers detected the demagnetization of the motor by conducting motor current signal analysis (MCSA), as the changes of current can reflect the changes of inductive electromotive force in the case of a malfunction in PMSM.

As a non-invasive method, MCSA is a convenient to collect, free from additional sensors, and can realize real-time detection, so it is well used in detecting mechanical breakdown and electrical failure of PMSM [18]. In quite a few studies, methods, such as fast Fourier transform (FFT) and wavelet transform, were employed to extract the stator current amplitude and harmonic changes after a demagnetization fault to diagnose demagnetization fault [19,20]. The frequency spectra of the stator current in a failed PMSM were analyzed and it was found that the contents of 0.25 and 0.5 harmonics were relatively higher in the frequency spectrum caused by partial demagnetization, while the 0.75 harmonic took dominance in the eccentric fault spectrum. Therefore, the different harmonic content of stator current spectrum after eccentric fault and partial demagnetization fault can be used as an indicator for identifying and diagnosing the two faults [21]. However, this method fails to take into consideration the impact imposed by the structure of the motor on the content of characteristics harmonics in the current under the circumstances of partial demagnetization. Hence, this fault characteristic cannot be adopted to diagnose demagnetization of motor with integral number of stator slots. The stator current is analyzed by FFT, indicating that no fault harmonics would be generated in stator phase current after the partial demagnetization occurred in motor with integral number of slots. However, researchers find that fault harmonics do exist in the branch current, so the analysis of stator branch current can overcome the problem that the characteristic harmonics disappear after a partial demagnetization fault due to the structural influence in some permanent magnet motors [22]. For a stator current of the PMSM with concentrated winding, analysis is conducted on the relationship between the expected partial harmonics caused by demagnetization and the numbers of magnetic poles and coils per phase. It is observed that the components of expected harmonics caused by demagnetization were mainly decided by factors, such as the number of phase coils, the ratio of slot number, and pole number of the motor [23].

Unlike partial demagnetization, the uniform one would not undermine the internal symmetry of the PMSM and would not generate new harmonics. However, that uniform demagnetization fault of the PMSM can still be diagnosed by analyzing the stator phase current of the motor. The result showed that uniform demagnetization led to the increase of the amplitude of fundamental waves and odd harmonics of stator phase current [24].

The above-mentioned studies only focused on the diagnosis and mode recognition of demagnetization in PMSM, but failed to probe into the number of magnetic poles with demagnetization fault and the fault location. Demagnetization databases for each permanent magnet were set up based on the back electromotive force of a single detection coil. Moreover, the demagnetization permanent magnet can be positioned by comparing the correlation of the real-time back electromotive force demagnetizing database of the motor and distinguishing between the uniform demagnetizing fault and the local demagnetization fault. Further, the fault degree can be quantified [25]. However, it is noteworthy that the detection coil will increase the cost of the motor and disrupt the motor's symmetry. Therefore, in combination with researches on the harmonic changes of electrical signals in demagnetized motor, this paper tries to explore the real reason for the disappearance of the characteristic harmonics after the demagnetization fault of the permanent magnet motor with a specific pole-to-slot ratio from the perspective of the motor branch [19,20]. Meanwhile, unlike phase current, the stator branch current was adequate in terms of amount [25] to be analyzed in terms of time-domain and frequency-domain information in the demagnetized motor. Hence, after the analysis of the stator branch current, a non-invasive method is proposed, which can realize real-time diagnosis, mode recognition, and location determination of the demagnetization fault in the motor, as well as obtain the number of demagnetized permanent magnets. The demagnetization fault model of PMSM is studied through finite element software, and the accuracy of the proposed methodology is verified.

The main structure of this paper is as follows. In Section 2, an analytical model of demagnetization fault is established for analyzing how the demagnetization fault affects the inductive electromotive force and current of the armature in PMSM. In Section 3, the process of establishing the database underpinned by demagnetization samples, the calculation methods of relative coefficients, and the determination of the demagnetization fault threshold value are presented in detail. In Section 4, the procedures of identifying and locating the demagnetization fault in the PMSM are illustrated. In Section 5, the results are verified through the finite element method (FEM). A brief summary of this paper is presented in Section 6.

## 2. Analysis on Demagnetization Fault in PMSM

### 2.1. Analysis on Back-EMF in Slot

According to the armature reaction in the PM motor, the PM in the rotor generates back-EMF in the stator winding. When the motor is in normal operation, the Back-EMFs generated by PM on the rotor in each stator slot present periodical variations. When a partial demagnetization fault occurs in the motor, the back-EMF in a single slot is irregular, as shown in Figure 1. The reason behind such a phenomenon is that the magnetic density generated by the demagnetized permanent magnet would decrease, and when this permanent magnet acts on a particular slot, the generated Back-EMF is reduced while those of other PM are free from demagnetization and remain unchanged. Therefore, within a mechanical cycle, the Back-EMF generated in one stator slot is distorted.

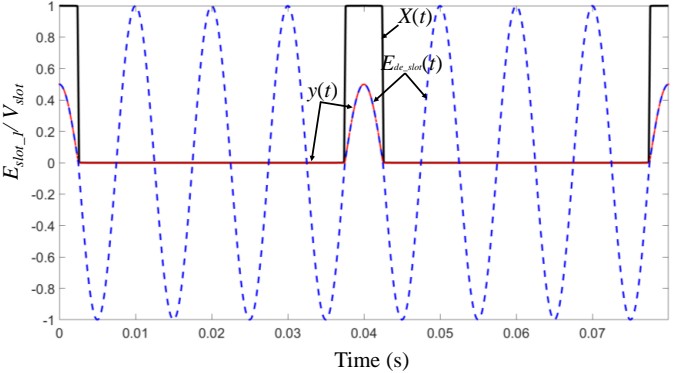

**Figure 1.** Effects of Simulated Demagnetization on Back−EMF.

In order to illustrate the above-mentioned process, the impacts of the geometric layout of the rotor magnets and the stator structure are not taken into account. The counter voltage generated by a healthy motor in a single slot presents sinusoidal variation. It is assumed that a certain permanent magnet of the motor suffered 50% of demagnetization, and then the Back-EMF generated by this magnet passing through the analysis slot would be halved. This effect is simulated in Figure 1 by making $E_{solt}$ minus $y(t)$.

In mathematics, the wave form $y(t)$ can be obtained by fundamental sine wave multiple square waves $x(t)$ and a multiple corresponding demagnetization coefficient. The frequency of square wave $x(t)$ is $f_e/p$ and the duty ratio is $d = 1/2p$. The square wave $x(t)$ can be expressed by Fourier series:

$$x(t) = \frac{1}{2p} + \sum_{n=1}^{\infty} \frac{2}{n\pi} \sin(\pi n d) \cos\left(\frac{2n\pi f_e t}{p}\right) \tag{1}$$

where $p$ is the number of pole pairs of the motor and $f_e$ is the fundamental frequency of the motor. Then, the wave form $y(t)$ can be expressed as:

$$y(t) = K_{de} \cos(2\pi f_e t) \left[ \frac{1}{2p} + \sum_{n=1}^{\infty} \frac{2}{n\pi} \sin(\pi n d) \cos\left(\frac{2n\pi f_e t}{p}\right) \right] \tag{2}$$

where $K_{de}$ represents the demagnetization severity of a single permanent magnet. Calculations are performed according to Equation (2), into which the $d = 1/2p$ is put.

$$y(t) = \frac{K_{de}}{2p} \cos(2\pi f_e t) K_{de} \sum_{n=1}^{\infty} \frac{1}{n\pi} \sin(\frac{\pi n}{2p}) \cos \underbrace{\left( 2\pi f_e t \left( 1 \pm \frac{n}{p} \right) \right)}_{\text{harmonic components}} \tag{3}$$

Then, in a fault motor, the back-EMF generated at one stator slot by the rotor of the demagnetized motor, namely $E_{de\_slot}$, can be expressed as:

$$E_{de\_slot} = V_{slot} \cos(2\pi f_e t) - V_{slot} y(t) \tag{4}$$

where $V_{slot}$ is the amplitude of the back-EMF generated by a normal motor in one stator slot. It is worth noting that the unit magnitude of all the variables below depend on $V_{slot}$ and are consistent with it. $E_{de\_slot}$ can be obtained by putting Equation (3) into Equation (4).

$$E_{de\_slot} = V_{slot} \left( 1 - \frac{K_{de}}{2p} \right) \cos(2\pi f_e t) - K_{de} V_{slot} \sum_{n=1}^{\infty} \frac{1}{n\pi} \sin(\frac{\pi n}{2p}) \cos \underbrace{\left( 2\pi f_e t \left( 1 \pm \frac{n}{p} \right) \right)}_{\text{harmonic components}} \tag{5}$$

It can be observed from the first part of Equation (5) that the amplitude of the back-EMF $E_{de\_slot\_1}$ is reduced due to the demagnetization of a single permanent magnet, and the new amplitude can be theoretically calculated by $1 - K_{de}/2p$. The second part of Equation (5) reveals that demagnetization leads to the generation of $(1 \pm n/p)$ harmonics in the back-EMF of a single slot.

### 2.2. Analysis on Back-EMF in Branch

According to the theory of AC motor winding, the motor windings are classified into series windings (all coil assemblies are in series connection to form a branch) and parallel windings (coil assemblies are in parallel connection and each assembly serves as an individual branch). Therefore, the analysis of branch back-EMF is of fundamental significance in analyzing phase back-EMF. In light of the superposition principle of Back-

EMF, in a normal motor, the branch Back-EMF is the addition of slot back-EMF of the branch, which can be expressed as:

$$E_b = 2Gk_{N1}E_{slot} \tag{6}$$

where $k_{N1}$ is the coefficient of motor winding; $G$ is the series coefficient of phase winding, which can be expressed in Equation (7); $E_{slot}$ is the back-EMF generated by the rotor of a normal motor in one stator slot. The phase back-EMF of the motor is equal to the branch back-EMF.

$$G = \begin{cases} \frac{p}{a}q & \text{single-layer winding} \\ 2\frac{p}{a}q & \text{double-layer winding} \end{cases} \tag{7}$$

where $q$ is the number of slots per pole per phase and $q = Q/2mp$; $a$ is the number of the branch of the motor.

In a motor with integral number of slots, the branch windings are symmetrically distributed in the mechanical circular. When one of the permanent magnets in this motor is demagnetized, the back-EMF of each branch in the motor can be differentially affected within a mechanical cycle. A motor with three parallel branches was taken as an example, and the Back-EMF of each branch in the fault motor can be expressed as follows:

Back-EMF of the first branch $E_{de\_b1}$ is expressed as:

$$\begin{aligned} E_{de\_b1} &= GK_{N1}V_{slot}\left(1 - \frac{K_{de}}{2p}\right)\cos(2\pi f_e t) \\ &- K_{de}GK_{N1}V_{slot}\sum_{n=1}^{\infty}\frac{1}{n\pi}\sin(\frac{\pi n}{2p})\cos\underbrace{\left(2\pi f_e t\left(1 \pm \frac{n}{p}\right)\right)}_{\text{harmonic components}} \end{aligned} \tag{8}$$

Back-EMF of the second branch $E_{de\_b2}$ is expressed as:

$$\begin{aligned} E_{de\_b2} &= GK_{N1}V_{slot}\left(1 - \frac{K_{de}}{2p}\right)\cos(2\pi f_e t) \\ &- K_{de}GK_{N1}V_{slot}\sum_{n=1}^{\infty}\frac{1}{n\pi}\sin(\frac{\pi n}{2p})\cos\underbrace{\left(2\pi f_e t\left(1 \pm \frac{n}{p}\right) + \frac{2\pi}{3}\right)}_{\text{harmonic components}} \end{aligned} \tag{9}$$

Back-EMF of the first branch $E_{de\_b3}$ is expressed as:

$$\begin{aligned} E_{de\_b3} &= GK_{N1}V_{slot}\left(1 - \frac{K_{de}}{2p}\right)\cos(2\pi f_e t) \\ &- K_{de}GK_{N1}V_{slot}\sum_{n=1}^{\infty}\frac{1}{n\pi}\sin(\frac{\pi n}{2p})\cos\underbrace{\left(2\pi f_e t\left(1 \pm \frac{n}{p}\right) + \frac{4\pi}{3}\right)}_{\text{harmonic components}} \end{aligned} \tag{10}$$

### 2.3. Analysis on Back-EMF in Phase

If the motor is of series winding, the phase back-EMF is equal to the vector sum of the above three branches, and the superposition of malfunction parts in the three branches can be expressed in Figure 2 Then, the phase back-EMF can be expressed as:

$$E_{de\_p} = 3GK_{N1}V_{slot}\left(1 - \frac{K_{de}}{2p}\right)\cos(2\pi f_e t) \tag{11}$$

It can be seen from Equation (11) that in phase back-EMF and in stator phase current, the fault harmonic components would disappear. For a motor with an integral number of slots and of series winding, demagnetization reduces the back-EMF of the stator and the amplitude of fundamental wave in phase current, but there would be no fault harmonic component.

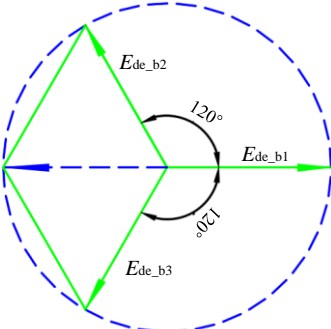

**Figure 2.** Vector Sum of Branch Back−EMF.

The superposition of back-EMF of the above branches is also applied to the motor of parallel winding. As shown in Figure 3, the circuit was equivalently processed by the Thevenin equivalent circuit, and the equivalent phase Back-EMF can be expressed as:

$$E_{Th} = GK_{N1}V_{slot}\left(1 - \frac{K_{de}}{2p}\right)\cos(2\pi f_e t) \tag{12}$$

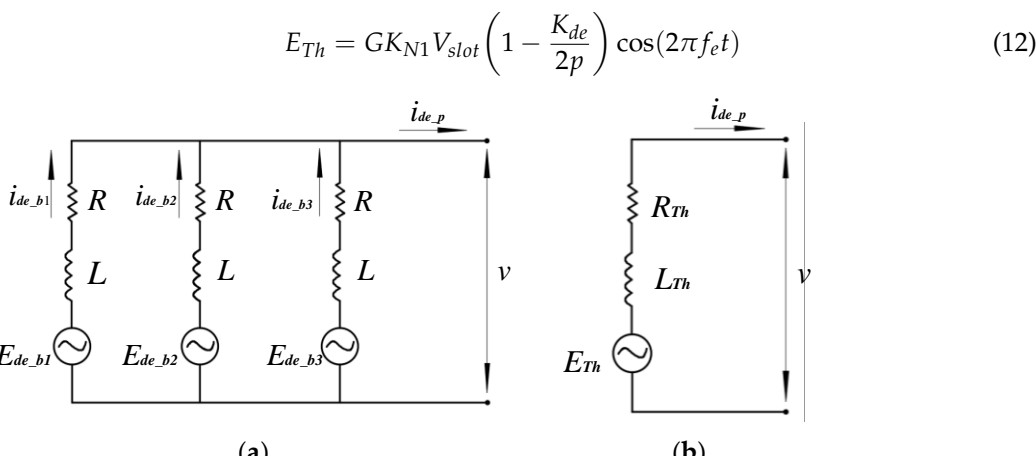

**Figure 3.** Thevenin Phase Equivalent Circuit: Parallel Circuit (**a**), Series Circuit (**b**).

It can be seen from Equation (12) that when partial demagnetization fault occurs in the permanent magnet motor of parallel winding, the fault harmonic components of phase back-EMF would disappear. This effect was also reflected in the stator phase current, in that fault harmonic components can be observed only in back-EMF and the current of the branch.

Therefore, for a permanent magnet motor with an integral number of slots and of symmetric winding, when partial demagnetization occurred, the fault characteristic harmonics in phase Back-EMF and in current would offset each other, so there is no fault harmonic component to observe. In other words, there would be no harmonics when the ratio of slot number and pole number was 3/2 (and its integral multiples), and fault components would only exist in the electrical signals of the branch, which is consistent with the conclusions drawn by Julio-César [15] and Cristian [22]. Therefore, in this paper, the branch current of the motor is selected to analyze the partial demagnetization fault of the permanent magnet motor.

### 2.4. Analysis on Branch Current

During the practical working of the motor, the odd harmonic components and phase positions of the actual motor current mattered. Therefore, their influence on the branch current should also be considered in the analysis of the branch current. The stator branch current of a normal motor $i_{h\_b}(t)$ and that of the fault motor $i_{de\_b}(t)$ can be expressed as:

$$i_{h\_b}(t) = \sum_{n=1}^{\infty} I_{h\_b}\cos(2\pi f_e t + \varphi_h) \tag{13}$$

$$i_{de\_b}(t) = \sum_{n=1}^{\infty} \left( I_{h\_b} \cos(2\pi f_e t + \varphi_h) - I_{de\_b} \underbrace{\cos\left(2\pi f_e t(1 \pm \frac{n}{p}) + \varphi_{de}\right)}_{\text{harmonic components}} \right) \qquad (14)$$

where $I_{h\_b}$ and $I_{de\_b}$ refer to the amplitude of stator branch current in normal motor and in fault motor, respectively; $\varphi_h$ and $\varphi_{de}$ represent the phase position of stator branch current in normal and fault parts of the motor, respectively. In this paper, residual values of the branch current in the normal motor and in the motor with partial demagnetization fault are taken as the signal for effective analysis of the current changes caused by the fault. The residual value of the branch current can be expressed as:

$$i_{residual}(t) = i_{h\_b}(t) - i_{de\_b}(t) = \sum_{n=1}^{\infty} I_{de} \underbrace{\cos\left(2\pi f_e t(1 \pm \frac{n}{p}) + \varphi_{de}\right)}_{\text{harmonic components}} \qquad (15)$$

It can be seen from Equation (15) that when a partial demagnetization fault occurs, there will be fractional fault harmonics in the stator branch current. However, the number and the location of fault permanent magnets cannot be determined by only analyzing the frequency spectra of the stator current. In order to facilitate the fault location, this paper proposes a methodology based on the stator branch current fault sample database referring to the literature [25]. First, a sample database, which is underpinned by residual values of branch current in permanent magnets that demagnetized in an orderly manner of the motor, is established. Second, the real-time residual values of the branch current are compared with those in the aforementioned sample database, and the Pearson correlation coefficients of the signal and the sample database are calculated. Third, the threshold value of fault is set to determine the number and location of fault permanent magnet.

## 3. Establishment and Application of Sample Database

### 3.1. Establishment of Sample Database

First, the finite element model of the motor with full load under rated and healthy conditions was simulated and analyzed to obtain signals of branch current within two mechanical cycles. Then, by arranging different permanent magnets in sequence to demagnetize, the signals of the branch current under the same working conditions are obtained. According to Equation (16), the residual values of branch current in each numbered permanent magnet with demagnetization fault were calculated and then numbered as: $a = \{[a]_1, [a]_2, \ldots, [a]_{2p}\}$.

$$[a]_j = i_{h\_b}(t) - i_{de\_b}^j(t) \quad j = 1, \ldots, 2p \qquad (16)$$

where $i_{de\_b}^j$ refers to the branch current of demagnetized magnetic pole whose serial number is j. Finally, the residual values of branch current (a) are standardized through Equation (17) to obtain the characteristics sample database of branch current in each demagnetized permanent magnet under rated conditions: $b = \{[b]_1, [b]_2, \ldots, [b]_{2p}\}$. Figure 4 shows how this sample database for No.1 permanent magnet is established, which is a similar approach to other demagnetization faults of permanent magnets.

$$[b]_j = \frac{[a]_j}{\max\left|[a]_j\right|} \quad j = 1, \ldots, 2p \qquad (17)$$

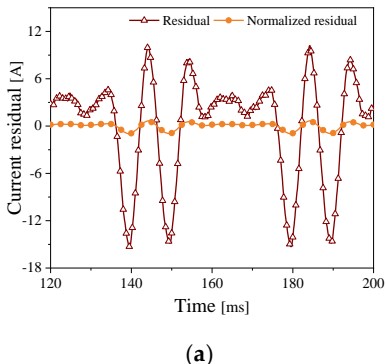
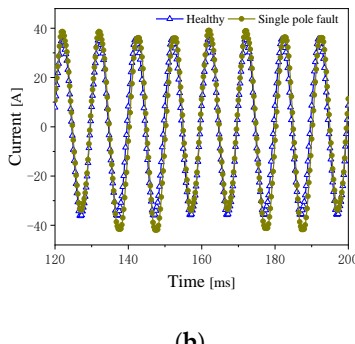

(**a**)　　　　　　　　　　　　　　　　　(**b**)

**Figure 4.** Establishment of Characteristic Sample Database; (**a**) Residual Values; (**b**) Standardization.

The establishment process of the characteristic sample database is as follows: First, a specific position of the motor is selected as the starting point, and two cycles of branch current signals in the healthy motor and the fault motor are intercepted respectively, as shown in Figure 4a. Secondly, the healthy current is minus with fault current to obtain the residual value of branch current. Finally, the residual current is standardized, as shown in Figure 4b.

### 3.2. Correlation Analysis

The branch current presents different wave forms as different permanent magnets of the motor are demagnetized. The residual values of the branch current during the actual operation of the motor are standardized to obtain a data group of $e$. This data group is compared with the sample database to calculate the Pearson correlation coefficient of each data group, so as to determine the location of the fault permanent magnet. The Pearson correlation coefficient $r_j$ can be calculated as:

$$r_j = \frac{\sum\limits_{k=1}^{n} (e_k - \bar{e})\left(b_{jk} - \overline{b_j}\right)}{\sqrt{\sum\limits_{k=1}^{n} (e_k - \bar{e})^2}\sqrt{\sum\limits_{k=1}^{n} \left(b_{jk} - \overline{b_j}\right)^2}} \tag{18}$$

where $n$ represents the data amount collected by signals; $k$ represents No. $k$ data point, and $k = 1, 2, \ldots, n$; $\bar{e}$ represents the average value of $e$ (standardized residual values); $\bar{b}$ represents the average value of each data group $[b]_j$ in the sample database. The Pearson correlation coefficient $r_j$ represents the correlation between two data groups. Hence, the larger the correlation number of the two sets of data is, the higher the similarity. The permanent magnet with high correlation indicates the location of demagnetization during the practical operation of the motor. However, given that several permanent magnets may simultaneously suffer demagnetization when a partial demagnetization fault in the motor occurs. Therefore, the number as well as the location of demagnetized permanent magnets should be figured out.

Prior to diagnosis of the motor, the threshold value of the demagnetization fault should be determined. The permanent magnet fault inevitably affects a certain part of the wave form of the branch current within a cycle, while the effect of every demagnetized permanent magnet on the branch current can be reflected in the wave form of the branch current in a uniform demagnetization fault. From Equation (18), one can obtain the Pearson correlation coefficient $r_j$ between the sample database and the standardized residual value group $e$ of branch current in uniform demagnetization fault. Under such circumstances, correlation coefficients were equal to each other, indicating the lowest correlation of demagnetization after the motor demagnetization fault, so the average value of these correlation coefficients can be set as the minimum fault threshold value. In the diagnosis and location of a fault in the motor, when a certain correlation coefficient $r_j$ is smaller than the threshold value, this indicates that the permanent magnet at this location does not suffer irreversible

demagnetization; when a certain correlation coefficient $r_j$ is greater than the threshold value, this indicates that the permanent magnet at this location suffer from demagnetization fault. The threshold value can be calculated according to the following equation:

$$r_{Th} = \frac{\sum\limits_{j=1}^{2p} r_j}{2p} \tag{19}$$

## 4. Diagnosis and Location of Demagnetization Fault

Based on the above-mentioned analysis, when the PMSM is under normal operation, the wave forms of phase current and branch current only contained fundamental waves and odd harmonics. When the motor demagnetization fault occurs, motors with different combinations of slot and pole showed different fault characteristics. For the permanent magnet motor with the slot-pole ratio of 3/2 and its integral multiples, the occurrence of a partial demagnetization fault will generate fault harmonics in the branch current rather than the phase current. Therefore, the motor fault can be diagnosed by the waveform change of the branch current.

The demagnetization of various PMs inevitably resulted in the occurrence of harmonics in the branch current, so it is difficult to locate the fault of the motor only by analyzing the frequency-domain information of the current and observing the content of harmonics. Within a cycle of the motor current, changes in the wave form of the current caused by the fault are different. As a consequence, faults in the motor can be located by comparing the actual motor current waveform with each partial demagnetization fault one in terms of correlation coefficients. The specific diagnosis and location of demagnetization fault in PMSM is shown in Figure 5. It should be noted that the diagnosis methodology in this paper is proposed under the pretext that the working conditions of the motor remained unchanged.

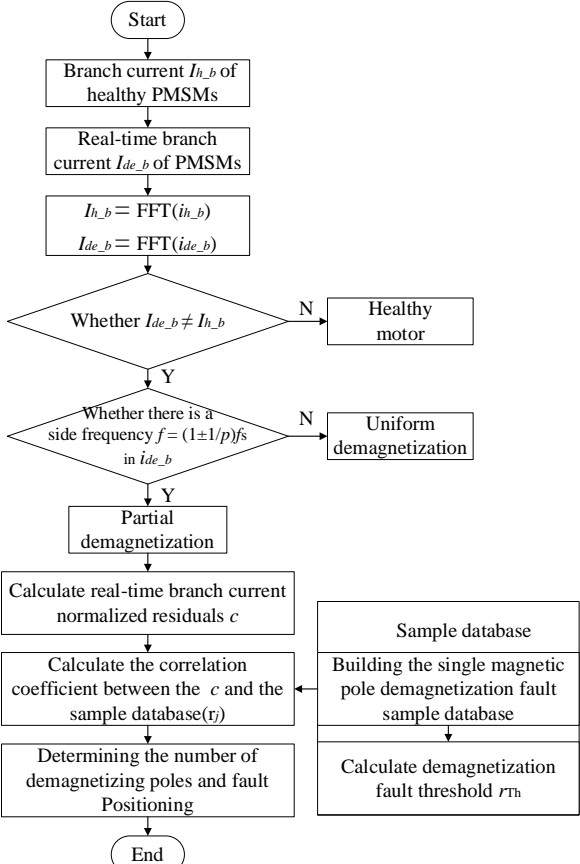

**Figure 5.** Flow Chart of Diagnosis and Location of Demagnetization Fault.

## 5. Simulation Analysis

### 5.1. Structure and Parameters of PMSM

In order to verify the aforementioned conclusions, a surface-mounted PMSM with 8 poles and 48 slots is taken as an example in this paper, and the full-load operation status under the rated speed of the motor was simulated. A two-dimensional analysis model is established in the finite element software to simulate models of the motor under healthy conditions, of the motor with partial demagnetization, and of the all-permanent magnet demagnetized motors with simultaneous demagnetization. The main structural parameters and models of the motor are shown in Table 1 and Figure 6. Meanwhile, the permanent magnets were serially numbered so that they can be distinguished from each other when establishing the sample database.

**Table 1.** Parameters of the adopted motor.

| Parameter | Value | Parameter | Value |
| --- | --- | --- | --- |
| Rated output power | 28.3 kW | Number of poles (2p) | 8 |
| Rated speed | 1500 r/min | Number of slots (Q) | 48 |
| Rated voltage | 345 V | Number of phase (m) | 3 |
| Length | 200 mm | Parallel branches (a) | 2 |
| Stator outer diameter | 230 mm | Magnet thickness | 4.5 mm |
| Stator inner diameter | 149 mm | Embrace | 0.87 |
| Rotor outer diameter | 147 mm | Magnet type | XG196/96 |
| Rotor inner diameter | 60 mm | | |

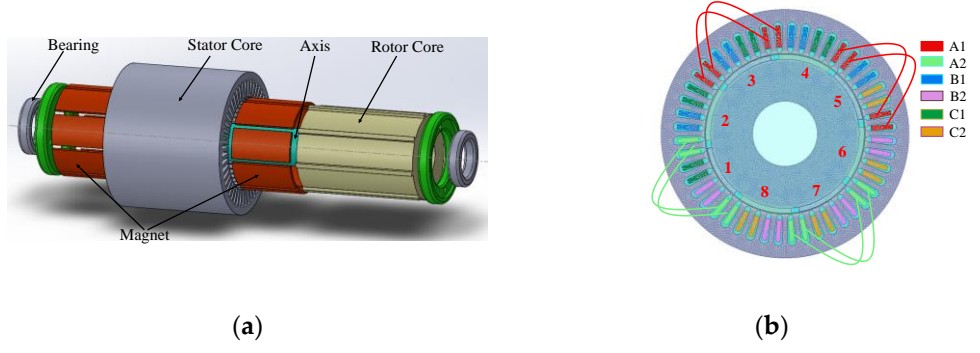

(**a**)                                                                                     (**b**)

**Figure 6.** Topology structure of the PMSM. (**a**) Geometric model. (**b**) Finite element model.

As shown in Figure 6, the permanent magnet motor simulated in this paper has two parallel branches (*a* = 2), and these two branches are symmetrically distributed in space, with each branch consisting four coil assemblies which are in series connection. In this paper, the branch current of A1 is collected for fault analysis of the motor.

### 5.2. Diagnosis and Mode Recognition of Demagnetization Fault

In order to verify the accuracy of the above analysis, the demagnetization of No.1, No. 2, and all permanent magnets are simulated in this paper, respectively, so as to further analyze the influence of partial demagnetization and uniform demagnetization on the stator phase current and the branch current. The simulated demagnetization model is established by setting the remanence Br of the permanent magnet material. During the establishment of the model, different permanent magnets are set to be demagnetized in a sequential manner for establishing sample database of demagnetization fault. The degree of demagnetization of each permanent magnet remains the same. The severity of demagnetization in each permanent magnet is 50% in this paper.

As shown in Figure 7a, when the No. 1 permanent magnet suffered an irreversible demagnetization fault, the amplitude of A phase current increased, but there are no side-frequency harmonics, so the phase current cannot indicate the existence of a partial demagnetization signal. As shown in Figure 7b, in the frequency spectrum of the branch current,

the demagnetization fault not only resulted in the increase of amplitude, but also resulted in obvious side-frequency components. Consistent with the above analysis, due to the symmetry of the motor, the fault harmonics caused by the partial demagnetization offset each other in the two-branch current of phase A. Therefore, the partial demagnetization fault of the motor can be recognized by analyzing the stator branch current.

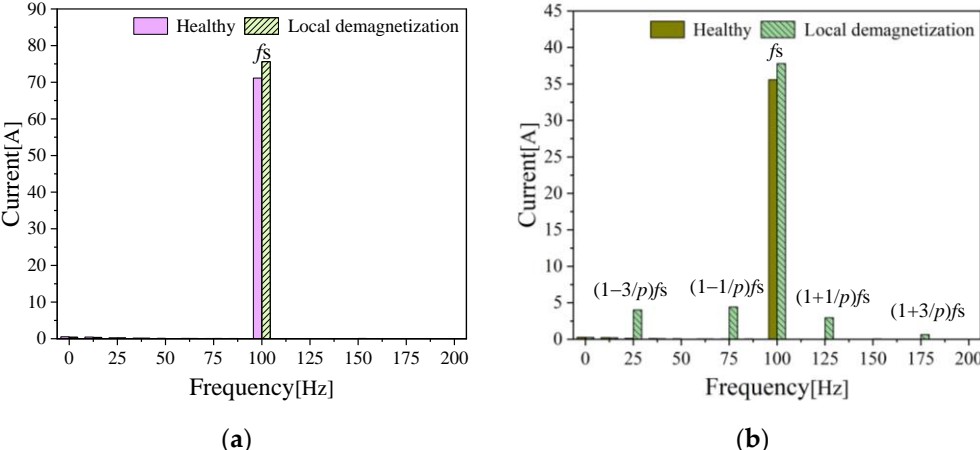

(**a**)  (**b**)

**Figure 7.** Frequency Spectra of Phase Current (**a**) and Branch Current (**b**) in Permanent Magnet with Local Demagnetization.

Figure 8 represents the frequency spectra of current of all permanent magnets simultaneously suffering 50% of demagnetization in the PMSM. It can be seen that the fundamental amplitudes of both the phase current and the branch current in the fault motor increased, but there are no harmonic components. Hence, it can be concluded that there is a uniform demagnetization fault in the motor if the current amplitude is increased while no harmonic components existed.

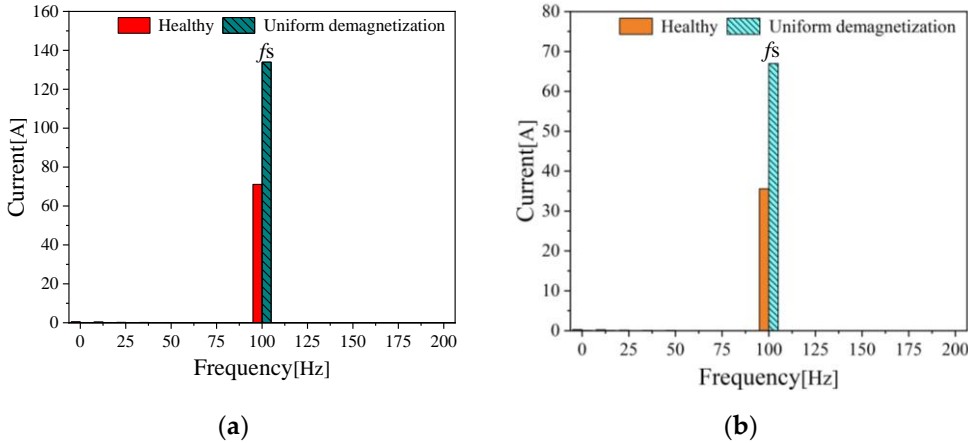

(**a**)  (**b**)

**Figure 8.** Frequency Spectra of Phase Current (**a**) and Branch Current (**b**) in Permanent Magnet with Uniform Demagnetization.

Figure 9 compares the demagnetization branch current of permanent magnet No.1 and No.2. It can be observed that if the location of demagnetized permanent magnets differed, the changes of the current in time-domain also varied (as shown in Figure 9a). However, the fault characteristics reflected in frequency-domain are consistent (as shown in Figure 9b). Therefore, no matter whether partial demagnetization fault occurs on any certain permanent magnet, the analysis of the fault harmonic component of the branch current can accurately identify the demagnetization fault. However, analysis of the frequency spectra of the branch current alone is not adequate to locate a demagnetization magnet in the motor, so the time-domain information of the branch current is required for analysis.

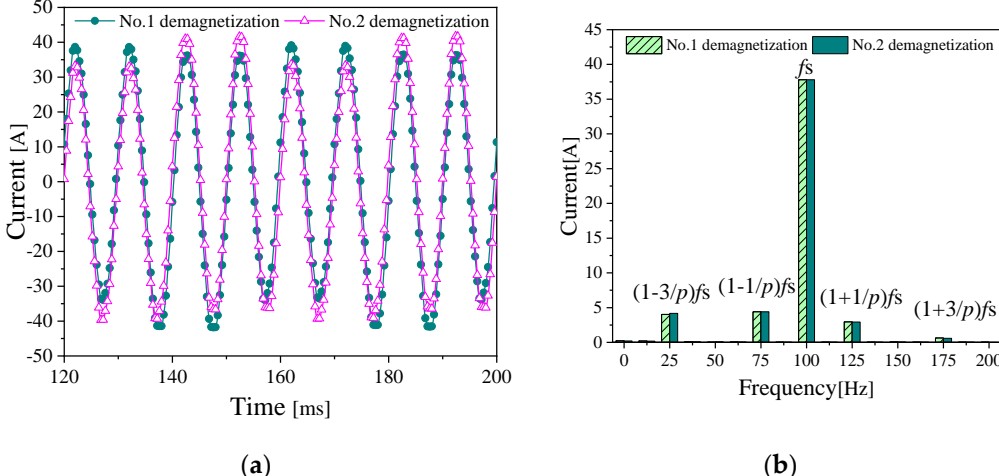

**Figure 9.** Time−Domain Analysis (**a**) and Frequency−Domain Analysis (**b**) of the Branch Current in Permanent Magnet No. 1 and No. 2 Demagnetized.

As shown in Figure 9a, the waveform changes of the branch current in different demagnetized permanent magnets varied. The reason for this is that when the location of demagnetized permanent magnet changed, the time of the demagnetized magnet acted on the A1 branch winding also varied, so the branch current changes differently in one mechanical cycle. The residual value between the branch current of the fault motor and that of the healthy motor can reflect the changes of the branch current in the fault motor. Therefore, the location of the demagnetization fault can be determined by establishing a sample database consisting of the residual value of the branch current in each demagnetized permanent magnet and then comparing the waveform of the branch current residual value of the motor working in real time with the sample database. Besides, according to the steps of establishing the fault sample database, the demagnetization models with 8 permanent magnets are numerically studied by means of FEM to obtain the residual values of the branch current. These residual values are normalized to obtain the fault sample database containing 8 groups of the motor data.

For the purpose to determine the fault threshold value of the simulated motor, it is of need to analyze the correlation coefficient $r_j$ between the standardized residual values e of the branch current in the motor with uniform demagnetization and the fault sample database. As different severity of uniform demagnetization may affect threshold value, five models based on different severity of uniform demagnetization are analyzed to ensure the accuracy of the threshold value. The correlation coefficients $r_j$ are shown in Table 2.

**Table 2.** Correlation coefficients between uniform demagnetization fault and fault sample database.

| Sample Characteristic Quantity Numbered | Uniform Demagnetization Fault Degree | | | | | | | | |
|---|---|---|---|---|---|---|---|---|---|
| | Fault 10% | Fault 20% | Fault 30% | Fault 40% | Fault 50% | Fault 60% | Fault 70% | Fault 80% | Fault 90% |
| 1 | 0.6965 | 0.6946 | 0.6956 | 0.695 | 0.6962 | 0.695 | 0.6946 | 0.6939 | 0.6927 |
| 2 | 0.6969 | 0.6965 | 0.697 | 0.697 | 0.6967 | 0.6968 | 0.6953 | 0.6959 | 0.6951 |
| 3 | 0.6963 | 0.696 | 0.6962 | 0.696 | 0.6949 | 0.6955 | 0.6951 | 0.6942 | 0.6932 |
| 4 | 0.6973 | 0.696 | 0.6972 | 0.697 | 0.6968 | 0.6967 | 0.6963 | 0.6956 | 0.6943 |
| 5 | 0.6964 | 0.6964 | 0.6964 | 0.6964 | 0.6952 | 0.696 | 0.6957 | 0.695 | 0.6937 |
| 6 | 0.6973 | 0.6959 | 0.6971 | 0.697 | 0.6969 | 0.6964 | 0.6969 | 0.6951 | 0.6934 |
| 7 | 0.695 | 0.6945 | 0.695 | 0.6945 | 0.6958 | 0.6948 | 0.6944 | 0.6938 | 0.6925 |
| 8 | 0.697 | 0.6962 | 0.697 | 0.6969 | 0.6868 | 0.6966 | 0.6962 | 0.6955 | 0.6943 |
| Average | 0.6965 | 0.6958 | 0.6964 | 0.6964 | 0.6962 | 0.696 | 0.6956 | 0.6949 | 0.6937 |

Table 2 shows the correlation coefficients between the branch current of the PMSMs with the fault sample database for uniform demagnetization levels from 10% to 90%. The minimum value of these coefficients is 0.6925, and the average value is greater than 0.69 The consistency of these data serves as a prove that the threshold value is free from the influence of different severity of uniform demagnetization, and the threshold value can accurately reflect whether there is demagnetization fault in the PMSM. The set threshold must be smaller than the correlation coefficient of all uniform demagnetization. When the PMSM is uniformly demagnetized, it can be determined that all permanent magnets are irreversibly demagnetized. Considering the influence of noise signals and numerical simulation errors, the fault threshold value of the motor could be set to 0.69. If the threshold value is greater than the threshold value of 0.69, then it can be determined that the permanent magnet in this position suffered demagnetization. The number of permanent magnets greater than the threshold value is exactly the number of permanent magnets with demagnetization fault.

### 5.3. Determination of the Number and the Location of Demagnetized Permanent Magnet

To figure out whether the method proposed in this paper to determine the number and the location of demagnetized PMSM is correct or not, simulations were conducted with the motor by setting different types and numbers of faults in permanent magnets. The starting point of data collection is consistent with that of the fault sample database.

The demagnetization severity of the test samples in this paper is different from that of the fault sample database for the purpose of distinction. The obtained signals of branch current in the motor are processed in accordance with the aforementioned methods to obtain residual values $e$. The residual values are then standardized to obtain the real-time working signals of the motor. Figure 10 shows the correlation coefficients $r_j$ between the demagnetization signal groups e of No.1–No.8 permanent magnets in the test set and $b_j$ from the fault sample database.

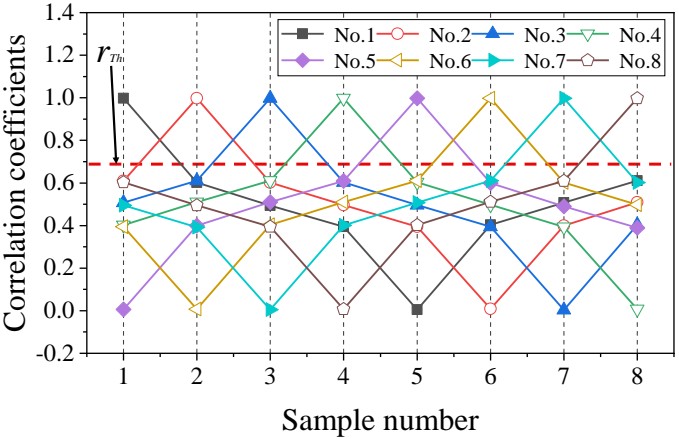

**Figure 10.** Correlation Coefficients of Single Demagnetized Permanent Magnets in the Test Set.

As shown in Figure 10, there is only one point where the correlation coefficient of each group of data in the test set exceeds the threshold, indicating that there is an occurrence of irreversible demagnetization, and the number of demagnetized permanent magnet is 1. Meanwhile, the serial number of the permanent magnet that is considered to be faulty is consistent with the serial number of the demagnetized permanent magnet set in the test set. Therefore, the location of demagnetized magnet can be accurately determined by comparing the correlation coefficients, which is consistent with the above-mentioned analysis.

To further verify the accuracy of the method proposed in this paper, we tested the correlation coefficients $r_j$ between the demagnetization signal data groups $e$ of permanent magnets with different number and location and $b_j$ from the fault sample database. The results are shown in Figure 11.

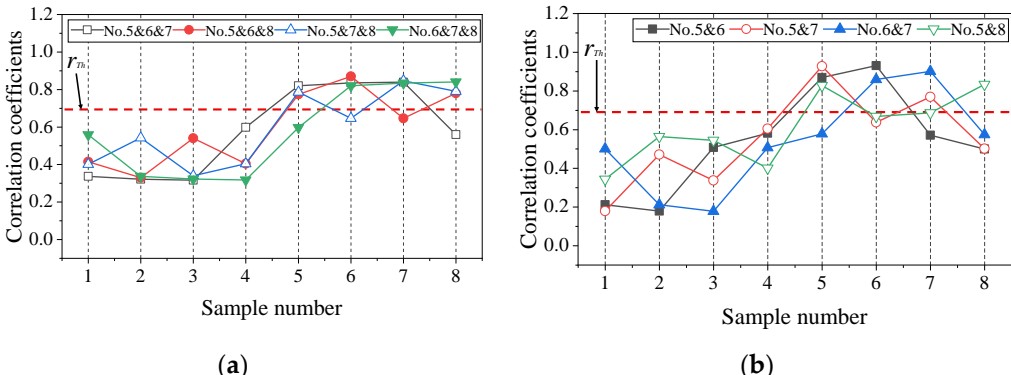

(**a**)            (**b**)

**Figure 11.** Correlation Coefficients of demagnetized Magnets in Different Locations of the Test Set: Two Demagnetized Permanent Magnets (**a**); Three Demagnetized permanent Magnets (**b**).

As shown in Figure 11a, when two permanent magnets pole faults are set, there are two correlation coefficients $r_j$ of each data group in the test set greater than the threshold value. Hence, there are two points with irreversible demagnetization, and the location of these two points corresponded with the serial numbers of demagnetized permanent magnets in the test set. While in Figure 11b, 3 demagnetized permanent magnets are set based on the data in test set, and the number and location of demagnetized magnets in the PMSM can be accurately determined by means of the correlation coefficient $r_j$. In Table 3, three kinds of demagnetization are summarized in order to straightforwardly demonstrate that under these three kinds of demagnetization, the number and location of the demagnetized magnet poles can be determined by the correlation coefficient $r_j$ between the data group $e$ of demagnetization signals and the data $b_j$ from the fault sample database. In conclusion, the simulation results in this section well prove that the method proposed in this paper can effectively recognize the number of demagnetized permanent magnets and locate the positions.

**Table 3.** Correlation coefficients between local demagnetization fault and fault sample database.

| Sample Database | One-Pole Fault | | | | | | | |
|---|---|---|---|---|---|---|---|---|
| | 1 | 2 | 3 | 4 | 5 | 6 | 7 | 8 |
| 1 | **0.998** | 0.61 | 0.508 | 0.402 | 0.006 | 0.394 | 0.495 | 0.603 |
| 2 | 0.602 | **0.998** | 0.611 | 0.51 | 0.4 | 0.008 | 0.392 | 0.495 |
| 3 | 0.494 | 0.602 | **0.998** | 0.611 | 0.51 | 0.403 | 0.005 | 0.395 |
| 4 | 0.394 | 0.496 | 0.603 | **0.998** | 0.61 | 0.51 | 0.401 | 0.008 |
| 5 | 0.0052 | 0.394 | 0.496 | 0.603 | **0.998** | 0.611 | 0.507 | 0.402 |
| 6 | 0.403 | 0.009 | 0.395 | 0.497 | 0.6 | **0.998** | 0.611 | 0.511 |
| 7 | 0.508 | 0.4 | 0.004 | 0.393 | 0.49 | 0.602 | **0.998** | 0.61 |
| 8 | 0.611 | 0.509 | 0.403 | 0.008 | 0.39 | 0.497 | 0.602 | **0.998** |

| Sample Database | One-Pole Fault | | | | Three-Pole Fault | | | |
|---|---|---|---|---|---|---|---|---|
| | 5, 6 | 5, 7 | 6, 7 | 5, 8 | 5, 6, 7 | 5, 6, 8 | 5, 7, 8 | 6, 7, 8 |
| 1 | 0.211 | 0.179 | 0.5 | 0.343 | 0.337 | 0.415 | 0.401 | 0.559 |
| 2 | 0.179 | 0.471 | 0.211 | 0.564 | 0.322 | 0.33 | 0.542 | 0.337 |
| 3 | 0.508 | 0.337 | 0.178 | 0.544 | 0.317 | 0.54 | 0.34 | 0.323 |
| 4 | 0.581 | 0.605 | 0.507 | 0.4 | 0.599 | 0.404 | 0.404 | 0.318 |
| 5 | **0.869** | **0.927** | 0.578 | **0.827** | **0.82** | **0.774** | **0.786** | 0.597 |
| 6 | **0.93** | 0.637 | **0.859** | 0.668 | **0.835** | **0.87** | 0.646 | **0.82** |
| 7 | 0.571 | **0.769** | **0.901** | 0.686 | **0.84** | 0.647 | **0.847** | **0.834** |
| 8 | 0.499 | 0.502 | 0.574 | **0.833** | 0.56 | 0.78 | **0.791** | **0.841** |

Bold indicates an irreversible demagnetization fault of the permanent magnets at this location.

## 6. Discussion

In the past research, using the Back-EMF to locate the demagnetization fault of PMSM requires arranging the detection coil inside the motor. This not only increases the diagnostic cost, but also represents an invasive diagnostic method that is difficult to implement. It can be seen from the results of this paper that the demagnetization fault of the PMSM can be effectively detected by directly analyzing the correlation coefficient of the stator branch current signal. At the same time, this method can determine the number and location of PMSM demagnetization faults. We improved the intrusive fault location method as well as considered the influence of the slot-to-pole ratio of the PMSM on the detection and location of demagnetization faults. However, the method proposed in this paper only analyzes the operation of the motor under the condition of constant speed. Therefore, how to locate the faulty magnetic pole when the motor is running at variable speed could be focused on in the future works.

## 7. Conclusions

In this paper, with the varied topologies of the PMSM being taken into consideration, we analyzed the interpretation model of the Back-EMF and the stator current in the PMSM which suffered a demagnetization fault. On this basis, it also proposed methods for the diagnosis and mode recognition as well as the determination of the number and the location of a demagnetization fault of the PMSM. FEA is adopted to verify the feasibility of above-mentioned methods. Conclusions can be drawn as follows:

The uniform demagnetization fault results in the amplitude changes of the Back-EMF and the stator current. Because of the topological structure, when the PMSM suffered partial demagnetization, there are no fault harmonic components in the Back-EMF and the stator phase current of the motor, whose slot-pole-ratio was 3/2, and its integral multiples. Therefore, the stator current cannot be taken as an indicator for the detection of a partial demagnetization fault in such motors.

The uniform and partial demagnetization faults of the PMSM with all slot-pole ratios can be effectively diagnosed and recognized by analyzing the amplitude and the harmonic components of the stator branch current of the PMSM.

By calculating the correlation coefficient between the residual value of the normalized branch current and the sample database when the motor actually works, it is possible to compare the threshold value. The number of magnetic poles with demagnetization fault in the PMSM and location of the fault can be effectively determined.

**Author Contributions:** Conceptualization, Y.Y. and H.G.; methodology, Y.Y.; software, H.G.; writing–original draft, Y.Y. and H.G. writing—review and editing, Y.Y., Q.C., P.L. and S.N. All authors have read and agreed to the published version of the manuscript.

**Funding:** This research was supported in part by the National Natural Science Foundation of China (Grant No. 52067006, 52162044), in part by the Foreign expert Bureau of the Ministry of science and technology of China (Grant No. G2021022002L), in part by long-term project of innovative leading talents in the "Double Thousand Plan" of Jiangxi Province (jxsq2019101027), by the Key Research Program of Jiangxi Province (Grant No. 20212BBE51014).

**Institutional Review Board Statement:** Not applicable.

**Informed Consent Statement:** Not applicable.

**Conflicts of Interest:** The authors declare no conflict of interest.

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
