# Peer review of "Demagnetization Fault Detection and Location in PMSM Based on Correlation Coefficient of Branch Current Signals"

_energies, doi:10.3390/en15082952_

Round 1
Reviewer 1 Report
The Authors described studies concerning on demagnetization fault detection and location in PMSM based .
The manuscript could be published in Energies after major revision. Below, several aspects have mentioned, which should be corrected and some doubts should be explained.
- The Abstract should be improved and contains the most important results from the manuscript.
- The motivation of studies should be highlighted in the Introduction.
- All magnitudes in equations should be explained in the text.
- The Discussion is very poor. The Authors should compare their results to data fro the literature.
The manuscript treats on extremely interesting topic. Generally, the Authors did some work. The manuscript could be published in Energies after corrections mentioned above.
Author Response
1.The Abstract should be improved and contains the most important results from the manuscript.
Thank you for your valuable comments. The abstract has been expanded in response to your comments and now includes a more specific research process of the paper and the important conclusions of the revised paper.
2.The motivation of studies should be highlighted in the Introduction.
Nowadays, the advantages of permanent magnet synchronous motor and some defects, as well as the importance of the research on the diagnosis of demagnetization faults, are explained in the first paragraph of the abstract of the revised paper. At the same time, it supplements the significance of the research on the location and number of fault poles. The motivation for this study is highlighted.
3.All magnitudes in equations should be explained in the text.
All the magnitudes in the equations in this paper are consistent with the back EMF amplitude Vslot generated by a normal motor in a stator slot. We also have a special description in the revised paper.
4.The Discussion is very poor. The Authors should compare their results to data from the literature.
Already supplementing the discussion section in this article based on your suggestions, your comments are greatly appreciated.
Based on several of your comments, we have made improvements to the revised paper as well. and we've highlighted the improvements in red.

Reviewer 2 Report
This study claims the following conclusions:
- The stator current cannot be taken as an indicator for the detection of partial demagnetization fault in such motor.
- By analyzing the amplitude and the harmonic components of stator branch current of PMSM, the uniform and partial demagnetization faults of all slot-pole-ratio can be effectively diagnosed and recognized.
- The number of magnetic poles with demagnetization fault and the location of the fault can be determined by calculating the correlation coefficient between the residual value of the normalized branch current and the sample database.
My comments are as shown in follow:
- As mentioned in the abstract, the full name, back electromotive force, must be used when the author first refers to back-EMF.
- In the introduction, the author did not mention any PMSM fault detection methods that are based on machine learning or deep learning. There are also many high-quality articles on motor detection that the author did not mention. A good review is also one of the contributions of an article.
- In figure 11, there should be (a) and (b) two sub-figures, however, there is only one sub-figure in figure 11.
- When the author uses the acronym PMSM in the text, all permanent magnet synchronous motors mentioned in the text must use PMSM.
- The author claimed that setting the threshold to 0.7 can determine that the permanent magnet in this position suffered demagnetization. However, for example, how can the author confirm whether the motor with the coefficient of 0.6999 has demagnetization or not? The author's selection of the threshold value needs to be further confirmed. In my opinion, doing only five different demagnetization simulations is too little. Simulating all demagnetization levels from 1 to 100% can further confirm the rationality of the threshold set by the author.
- The failure analysis of this research is completed through simulation. If there is a real motor to confirm the author's theory, the quality of this research will be greatly improved.
Author Response
- As mentioned in the abstract, the full name, back electromotive force, must be used when the author first refers to back-EMF.
Thank you for your valuable comments. We have added the full name of back-EMF where it first appears in the revised paper.
- In the introduction, the author did not mention any PMSM fault detection methods that are based on machine learning or deep learning. There are also many high-quality articles on motor detection that the author did not mention. A good review is also one of the contributions of an article.
As you mentioned, many scholars have applied the method of machine learning to the fault detection of permanent magnet synchronous motor. However, advanced machine learning algorithms are not the focus of this paper, so the relevant research results are rarely mentioned. At the same time, we have supplemented the relevant research results of some scholars in the introduction.
- In figure 11, there should be (a) and (b) two sub-figures, however, there is only one sub-figure in figure 11.
As you mentioned, in Figure 11, there should be two pictures, and due to our negligence, one picture is missing. Now that we've added in the text, thank you very much for the reminder.
- When the author uses the acronym PMSM in the text, all permanent magnet synchronous motors mentioned in the text must use PMSM.
Thank you very much for your reminder, we have used the full name of the permanent magnet synchronous motor that first appeared in the paper, and all the permanent magnet synchronous motors mentioned later use PMSM.
- The author claimed that setting the threshold to 0.7 can determine that the permanent magnet in this position suffered demagnetization. However, for example, how can the author confirm whether the motor with the coefficient of 0.6999 has demagnetization or not? The author's selection of the threshold value needs to be further confirmed. In my opinion, doing only five different demagnetization simulations is too little. Simulating all demagnetization levels from 1 to 100% can further confirm the rationality of the threshold set by the author.
Due to the time-consuming finite element simulation, it is difficult to achieve every degree of demagnetization from 1% demagnetization to 100% demagnetization. At the same time, we have now increased to nine different demagnetization simulations. In determining the threshold, we can only represent all uniform demagnetization failures by these evenly distributed nine demagnetization cases.
Considering that the set threshold must be smaller than the correlation coefficient of all uniform demagnetization, when the permanent magnet synchronous motor is uniformly demagnetized, it can be determined that all permanent magnets are irreversibly demagnetized. Among these correlation coefficients, the minimum value is 0.6925. However, if the threshold is chosen too small, the poles that have not been irreversibly demagnetized will be considered demagnetized. For example, in the case of demagnetization of No. 5, 6, and 8 permanent magnets, the correlation coefficient of No. 7 permanent magnet is 0.674. So, to avoid errors, the threshold should also be as large as possible. Considering the effect of numerical simulation errors, we decided to set the threshold to 0.69. If the threshold value is greater than the threshold value of 0.69, it can be determined that the permanent magnet at that location is demagnetized. And the finite element simulation results of this paper prove that the threshold value of 0.69 in this paper is reasonable. It is worth noting that the demagnetization degree of the simulation test set in this paper is 70%, and whether the threshold of 0.69 is accurate in other cases of demagnetization degree is worth further study.
- The failure analysis of this research is completed through simulation. If there is a real motor to confirm the author's theory, the quality of this research will be greatly improved.
As mentioned earlier, the failure analysis in this paper is done through simulation. We need experiments to test our theories. But since we currently lack the corresponding experimental conditions, it is difficult to confirm our theory with a real motor. At this stage, what we propose is to propose a method for detecting and locating demagnetization faults of permanent magnet synchronous motors, and the correctness of the method is verified by finite element simulation. When experimental conditions are available in the next phase, we will use real motors to test our theory. thank you very much for your suggestion.
Based on some of your comments, we have also made improvements to the revised document. We've highlighted improvements in blue.

Round 2
Reviewer 1 Report
The Authors took into account al reviewers comments and improved the manuscript. It could be published in present version.
Reviewer 2 Report
The author has answered all my questions.